# Simulation-Based Study on Round Window Atresia by Using a Straight Cochlea Model with Compressible Perilymph

Wenjia Hong [1] and Yasushi Horii [2],*

1 Graduate School of Informatics, Kansai University, Takatsuki 569-1095, Japan; hwjh2654568@gmail.com
2 Faculty of Informatics, Kansai University, Takatsuki 569-1095, Japan
* Correspondence: horii@kansai-u.ac.jp; Tel.: +81-(72)690-2476

**Abstract:** The sound stimulus received by the pinna is transmitted to the oval window of the inner ear via the outer ear and middle ear. Assuming that the perilymph in the scala vestibuli and scala tympani is compressible, we report that the sound wave generated in the cochlea due to the vibration of the oval window can be expressed by the combination of even and odd symmetric sound wave modes. Based on this new approach, this paper studies the cause of hearing deterioration in the lower frequency region seen in round window atresia from the viewpoint of cochlear acoustics. Round window atresia is an auditory disease in which the round window is ossified and its movement is restricted. Using the finite element method, a round window atresia model was designed and the acoustic behavior of the round window was discussed corresponding to the level of disease. From this, we report that the healthy round window works as a free-end reflector to the incident sound waves, but it also works as a fixed-end reflector in the case of round window atresia. Next, we incorporated the round window atresia model into a cochlear model and performed a simulation in order to determine the acoustic aspects of the cochlea as a whole. The simulation results indicate that hearing deterioration occurs in a lower frequency range, which is also coincident with the clinical reports (hearing deterioration of approximately 10 to 20 dB below 4000 Hz). Finally, we explain that the cause of hearing deterioration due to round window atresia is considered to be the even sound wave mode enlarging due to the fixed-end reflection at the ossified round window, and, as a result, the odd sound wave mode that generates the Békésy's traveling wave on a basilar membrane is significantly weakened.

**Keywords:** auditory mechanism; cochlea; traveling wave theory; compressible perilymph; even and odd mode analysis; round window atresia

## 1. Introduction

Humans can hear sounds from 20 Hz to 20,000 Hz, with a huge dynamic range of 120 dB and a high frequency resolution of 0.5% [1]. Over the last 100 years, many researchers have tried to elucidate the mechanism of the human auditory system; V. Békésy discovered a specific wave propagation on a basilar membrane of the cochlea and proposed the traveling wave theory in 1960 [2]. According to his study, higher frequency sounds excite the traveling waves near the base of the cochlea, and lower frequency ones excite them near the apex. This fact indicates that our auditory system detects sounds as a spectrum analyzer [3].

Another important aspect of the auditory system is that each outer hair cell works as an amplifier to enlarge small sounds detected by the basilar membrane; that is, when a traveling wave stimulates the outer hair cells, mechano-electrical transduction (MET) channels located at the tip of the stereocilia accept abundant potassium ions contained in the endolymph to enter inside of the outer hair cells [4]. Then, the shape of the protein motor *Prestin* [5] distributed on the surface of the cell membrane changes, causing the outer hair cells to contract [6,7]. As the small initial displacement of the basilar membrane

generated by the sound stimuli causes a large contraction movement of the outer hair cells, this series of movements is called *cochlea amplifier* [8]. The contraction movement of the outer hair cells pulls down the tectorial membrane and stimulates the inner hair cells. As a result, a potassium ion is absorbed in the inner hair cells in a similar manner; then, the sound information is transmitted to the brain through the vestibulocochlear nerves and, finally, humans perceive the sound [9,10].

On the other hand, theoretical simulations of the cochlea have a long history. In the 1950s, equivalent circuit models were developed based on the distributed circuit theory. The scala vestibuli and scala tympani were expressed as a transmission line, and the basilar membrane was designed as a large number of series RLC circuits connected in parallel between the scala vestibuli and scala tympani [11,12]. Until that time, the model was designed based on the idea that the sound waves propagated in the cochlea. In fact, it was pointed out that the treatment of the distributed constant circuit was important at frequencies range of more than 7000 Hz [11–13].

However, the auditory system is a biological organ composed of complex structures and materials. In order to clarify its behavior, we need a flexible analysis method and designers' experiences to simplify the models without losing their essential functions. Under such a background, in recent years, finite element analysis using fluid mechanics has become the mainstream of cochlear analysis [14–22]. Among them, some papers treated perilymph as an incompressible medium, while others treated it as a compressible medium to allow sound waves to propagate in the cochlea. In addition, the acoustics of the round window have also been studied using the finite element method. It is well known that the active actuator of the middle-ear implant is coupled to the round window membrane in order to compensate for hearing loss in patients who suffer from middle-ear disorders. To evaluate the coupling condition between the human ear and the actuator, a finite element model was applied [23], and a coupling impedance model was designed [24].

This paper studies the detailed mechanism of round window atresia by using a straight cochlea model with compressible perilymph. In Section 2, basic equations of the fluid dynamics applicable to the compressible media are introduced. At the beginning, a healthy cochlea model is designed and simulated. Since the model has a symmetric architecture against the scala media, even and odd-mode sound waves can travel independently from the cochlea base to the apex. From the viewpoint of the even and odd-mode theory, we explain how the odd sound wave mode generates the traveling wave and how the even sound wave mode contributes to determining the amplitude of the traveling wave [25]. In Section 3, we introduce the unique symptoms of round window atresia, which is an auditory disease where the round window is ossified and its movement is restricted. We design an ossified round window model and explain a way to express the stages of round window atresia by changing the Young's modulus of the round window membrane. Additionally, we discuss how the reflection properties of the round window change according to the stage of the disease. Following that, we apply the ossified round window to the cochlea model and argue how round window atresia affects the cochlear acoustics as a whole. Finally, in Section 4, we emphasize that, when we analyze the acoustic aspects of the cochlea, the perilymph should be treated as a compressible medium because the sound wave is traveling in the perilymph as a compression wave.

## 2. Acoustics of Cochlea Based on Even and Odd-Mode Analysis

### 2.1. Fluid Equations for Compressible Media

As mentioned above, in recent years, in the study of cochlea mechanics, the analysis of fluid dynamics has seemed to be the mainstream. It is generally said in fluid dynamics that a fluid can be treated as incompressible when it flows at a speed of less than 0.3 Ma, where 1.0 Ma is the velocity of sound waves traveling in the medium. For example, if the velocity of the sound waves in the perilymph is assumed to be 1520 m/s, the medium can be treated as incompressible when the perilymph flows at a speed of less than 456 m/s. However, as a specific case, the perilymph of the cochlea should be treated as a compressible medium

when we study the acoustic aspects of the cochlea, because the sound waves propagate in the cochlea as a compression wave. Therefore, the fluid equations for compressible media were applied to the cochlea models in this paper. Actual simulations were carried out by using the commercial software COMSOL Multiphysics Ver. 5.3, including an add-on acoustic module and a structural mechanics module. For simulations, assuming that the perilymph is the compressible Newtonian fluid, the continuity equation and the Navier–Stokes equation for compressible media were used:

$$\frac{\partial \rho}{\partial t} + \nabla \cdot (\rho \mathbf{u}) = 0 \tag{1}$$

$$\rho \left( \frac{\partial \mathbf{u}}{\partial t} + (\mathbf{u} \cdot \nabla)\mathbf{u} \right) = -\nabla p + \nabla \cdot (\mu(\nabla \mathbf{u} + (\nabla \mathbf{u})^T) - \frac{2}{3}\mu(\nabla \cdot \mathbf{u})\mathbf{I} \tag{2}$$

where $\mathbf{u}$ stands for the fluid velocity, $\mathbf{I}$ for the identity tensor, $p$ for the fluid pressure, $\rho$ for the fluid density, $\mu$ for the viscosity, and $t$ for time. $\nabla$ is a differential operator defined by

$$\nabla = \frac{\partial}{\partial x}\boldsymbol{i} + \frac{\partial}{\partial y}\boldsymbol{j} + \frac{\partial}{\partial z}\boldsymbol{k} \tag{3}$$

On the other hand, the basilar membrane was treated as an elastic material calculated by

$$\mathbf{M}\frac{\partial^2 \mathbf{u}}{\partial t^2} + \mathbf{C}\frac{\partial \mathbf{u}}{\partial t} + \mathbf{K}\mathbf{u} = \mathbf{f}(t) \tag{4}$$

where $\mathbf{M}$ is the mass matrix, $\mathbf{C}$ is the damping matrix, $\mathbf{K}$ is the stiffness matrix, and $\mathbf{f}(t)$ is the external force applied to the material.

### 2.2. Modeling of the Cochlea

Figure 1 shows an illustration of a healthy cochlea model including compressible perilymph. The original spiral-shaped cochlea was unrolled and slightly tapered to simplify the simulation. The total length of the cochlea was assumed to be $L_c$ = 35 mm [26]. At the cochlea's base, the total width was $W_{bc}$ = 1.2 mm [27] and the total height was $H_{bc}$ = 2.5 mm, while, at the cochlea's apex, the total width was $W_{ac}$ = 0.7 mm and the total height was $H_{ac}$ = 1.5 mm. The scala media was sandwiched symmetrically by the scala vestibuli and scala tympani, and the basilar membrane was embedded in the scala media, whose dimensions were defined by a whole length of $L_m$ = 34 mm, a width of $W_{bm}$ = 100 μm, and a thickness of $H_{bm}$ = 30 μm at the base, and a width of $W_{am}$ = 500 μm and a thickness of $H_{am}$ = 10 μm at the apex [28,29]. The Young's modulus, Poisson ratio, and mass density of the basilar membrane were $E$ = 1 MPa, $v$ = 0.49, and $\rho$ = 1200 kg/m$^3$, respectively. These values were determined to satisfy Greenwood's tonotopy equation [3]. Additionally, a helicotrema with a diameter of $D_h$ = 0.65 mm was set at 34.675 mm from the cochlea's base so as to connect the scala vestibuli and scala tympani [28]. Then, the scala vestibuli, scala tympani, and helicotrema were filled with compressible perilymph with a viscosity of $\mu$ = 0.7027 mPa·s, a density of $\rho$ = 994.6 kg/m$^3$, a sound velocity of $c$ = 1520 m/s, and a temperature of $T$ = 36 degrees Celsius to allow the sound waves to propagate as a compression wave. On the other hand, to represent the air-filled environment of the middle ear cavity, an air space with a length of $L_a$ = 3.0 mm, a width of $W_a$ = 1.2 mm, and a height of $H_a$ = 1.235 mm was connected to the base end of the scala tympani. An elliptical round window with a width of $W_r$ = 1.12 mm, a height of $H_r$ = 0.69 mm, a thickness of $T_r$ = 70 μm, a Young's modulus of $E$ = 1 MPa, a Poisson's ratio of $v$ = 0.49, and a mass density of $\rho$ = 1200 kg/m$^3$ was modeled at the boundary between the air space and the scala tympani [27,30–32]. In addition, an ideal hard boundary condition, that is, the condition that the normal component of the particle velocity is always zero on the wall's surface, was applied to the walls in the model, except for the round window and basilar membrane, so that the sound wave reflected perfectly without losses on their surfaces. Detailed settings of the finite element method and PC specification for simulations are summarized in Table 1.

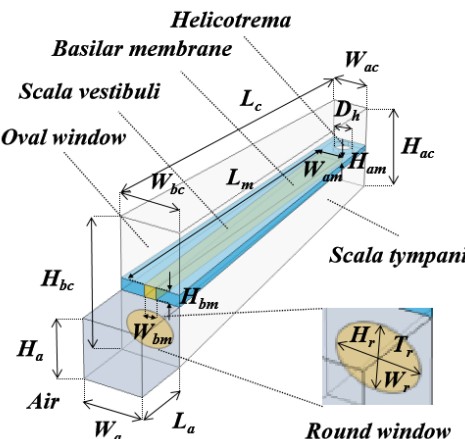

**Figure 1.** A straight-tapered cochlea model configured by scala vestibuli, scala media, and scala tympani. The scala vestibuli and scala tympani were connected by a helicotrema at the cochlea apex. The scala vestibuli, scala tympani, and helicotrema were filled with compressible perilymph, and a basilar membrane was embedded in the scala media. A continuous sinusoidal plane wave was excited at the oval window. The dimensions were $L_c$ = 35 mm, $W_{bc}$ = 1.2 mm, $H_{bc}$ = 2.5 mm, $W_{ac}$ = 0.7 mm, $H_{ac}$ = 1.5 mm, $L_m$ = 34 mm, $W_{bm}$ = 100 μm, $H_{bm}$ = 30 μm, $W_{am}$ = 500 μm, $H_{am}$ = 10 μm, $D_h$ = 0.65 mm, $L_a$ = 3 mm, $W_a$ = 1.2 mm, $H_a$ = 1.235 mm, $W_r$ = 1.12 mm, $H_r$ = 0.69 mm, and $T_r$ = 70 μm.

**Table 1.** Settings of the finite element method (FEM) simulations and PC specification.

| FEM Mesh Settings (typical cochlea model) | |
|---|---|
| Maximum mesh size | 1000 μm |
| Minimum mesh size | 10 μm |
| Mesh generation | automatic |
| Number of mesh elements | 1,033,262 |
| **FEM Settings for Time Domain Analysis** | |
| Time step | 0.01 ms |
| Time range | 0 ms – 40 ms |
| Computation time (depend on convergence) | 48 h – 72 h |
| **FEM Material Settings** | |
| Compressible perilymph | |
| Viscosity | 0.7027 mPa·s |
| Density | 994.6 kg/m³ |
| Sound velocity | 1520 m/s |
| Basilar membrane | |
| Density | 1200 kg/m³ |
| Young's modulus | 1 MPa |
| Poisson's ratio | 0.49 |
| Round window membrane | |
| Density | 1200 kg/m³ |
| Young's modulus | 1 MPa |
| Poisson's ratio | 0.49 |

**Table 1.** *Cont.*

| FEM Mesh Settings (typical cochlea model) | |
|---|---|
| **PC specification** | |
| CPU | Corei9–7980XE |
| Clock | 2.6 GHz |
| Memory | 128 GB |
| OS | Win 10 Pro 64bit |

### 2.3. Sound Waves in Cochlea and Generation of Traveling Waves

As shown in Figure 1, two sound observation lines were set so as to penetrate the center of the scala vestibuli and scala tympani, and the sound pressure levels were calculated on these lines when the continuous sinusoidal plane wave with a maximum pressure level of 1 Pa and a frequency of 5000 Hz was excited at the portion of the oval window. Then, the maximum swing (solid line, $t$ = 38.84 ms) and the minimum swing (dashed line, $t$ = 38.94 ms) of the sound pressures observed on these lines were plotted, as shown in Figure 2a. The upper graph shows the sound pressure in the scala vestibuli, and the lower one shows that in the scala tympani. The horizontal axis shows the position from the base to the apex in the cochlea. The actual sound pressure levels in the scala vestibuli and scala tympani varied between these lines with time.

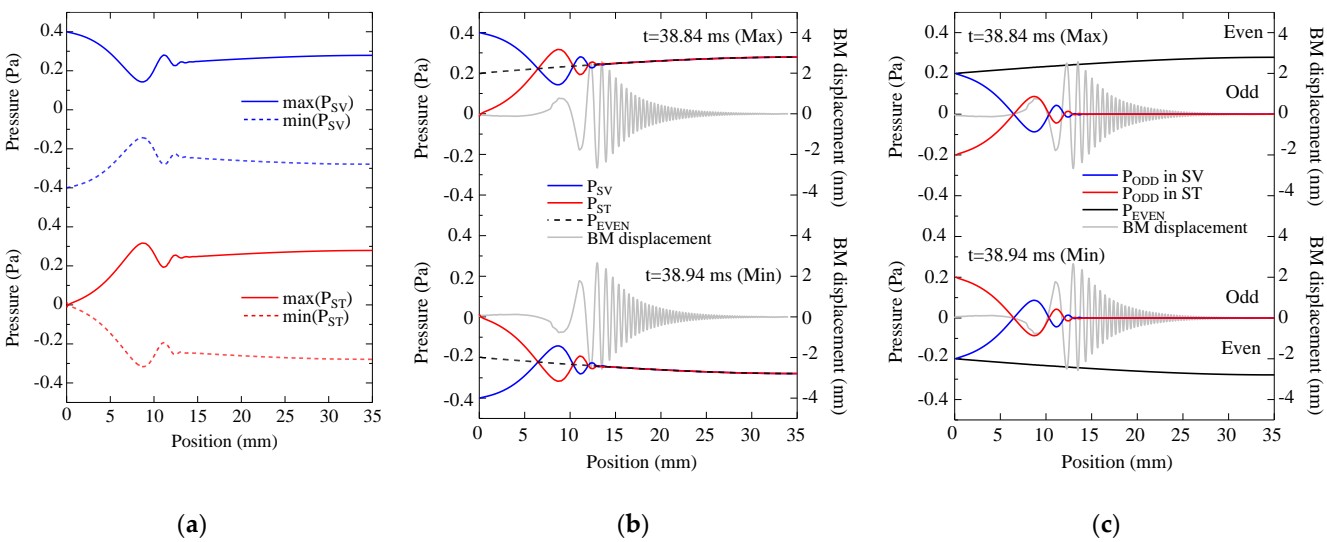

(**a**)                                                    (**b**)                                                    (**c**)

**Figure 2.** Evaluation of sound pressure levels in the scala vestibuli and scala tympani, and the displacement of the basilar membrane when a continuous sinusoidal plane wave with a pressure level of 1 Pa and a frequency of 5000 Hz was excited at the oval window. The horizontal axis shows the position in the cochlea. The observation times $t$ = 38.84 ms and 38.94 ms were chosen by trial and error so that the waveform swing $P_{EVEN}$ became the maximum or the minimum after the waveform reached a steady state. (**a**) Maximum (solid line, $t$ = 38.84 ms) and minimum (dashed line, $t$ = 38.94 ms) sound pressure levels in the scala vestibuli (in blue) and scala tympani (in red). (**b**) Sound pressure levels in the scala vestibule (in blue) and scala tympani (in red), and the displacement of the basilar membrane (in light gray). The upper and lower graphs present the simulation results observed at $t$ = 38.84 ms and $t$ = 38.94 ms, respectively. (**c**) Sound pressure levels of the even and odd modes, and the displacement of the basilar membrane (in light gray), observed at $t$ = 38.84 ms (the upper graph) and $t$ = 38.94 ms (the lower graph). The sound pressure levels of the odd modes in the scala vestibuli and scala tympani are shown in blue and in red, respectively.

Two interesting aspects can be read from the graph. The first one is that the sound pressure level did not swing at all and always became zero at the base of the scala tympani

(corresponding to the position of 0 mm in the cochlea), even though the sound pressure level at the base of the scala vestibuli swung largely from $-0.4$ Pa to $+0.4$ Pa with time. This is because the base of the scala tympani worked as a specific border between the air-filled outside region (an acoustic impedance of 440 Pa·s/m$^3$) and the perilymph-filled inside region (an acoustic impedance of 1.5 MPa·s/m$^3$) across the round window. In such a case, an acoustic wave traveling in the scala tympani heading to the round window was reflected mostly with a free-end reflection condition due to the large impedance mismatch between the air and the perilymph. As a result, the velocity of the medium became the maximum, and the pressure level became zero at the border.

The second interesting feature in Figure 2a is that the waveforms in the scala vestibuli and scala tympani vibrated largely only at the position from 0 mm to 13 mm. However, these vibrations were gradually reduced and vanished beyond 13 mm. Instead, the waveform became quite smooth from 13 mm to 35 mm. As explained in [25], these results imply that two types of sound waves were excited in the cochlea. In order to distinguish what kinds of sound waves overlapped in each scala, these waves were separated using the following procedure. Here, let us express the sound pressure levels on the observation lines as $P_{SV}(x, t)$ in the scala vestibuli and $P_{ST}(x, t)$ in the scala tympani, where $x$ and $t$ denote the position in the cochlea and the time variation, respectively. Then, we calculated:

$$P_{EVEN}(x,t) = \{P_{SV}(x,t) + P_{ST}(x,t)\}/2 \tag{5}$$

and

$$P_{ODD}(x,t) = P_{SV}(x,t) - P_{EVEN}(x,t) = \{P_{SV}(x,t) - P_{ST}(x,t)\}/2 \tag{6}$$

By using these expressions, the original $P_{SV}(x, t)$ and $P_{ST}(x, t)$ were expressed as follows.

$$P_{SV}(x,t) = P_{EVEN}(x,t) + P_{ODD}(x,t) \tag{7}$$

$$P_{ST}(x,t) = P_{EVEN}(x,t) - P_{ODD}(x,t) \tag{8}$$

The pressure levels of $P_{SV}(x, 38.84$ ms$)$ and $P_{ST}(x, 38.84$ ms$)$ are drawn in the upper region of Figure 2b in blue and red, respectively. In addition, the displacement of the basilar membrane was also calculated and presented in light gray in the same figure together with the $P_{EVEN}(x, 38.94$ ms$)$ in black. In the same way, the pressure levels of the $P_{SV}(x, 38.94$ ms$)$, $P_{ST}(x, 38.94$ ms$)$, and $P_{EVEN}(x, 38.94$ ms$)$, and the displacement of the basilar membrane, are presented in the lower region of the figure. Based on these results, the combination of the $P_{EVEN}(x, 38.84$ ms$)$ and $P_{ODD}(x, 38.84$ ms$)$, and that of the $P_{EVEN}(x, 38.94$ ms$)$ and $P_{ODD}(x, 38.94$ ms$)$, were calculated as expressed in Figure 2c. This result indicates that even and odd symmetric sound wave modes, $P_{EVEN}(x, t)$ and $P_{ODD}(x, t)$, respectively, exist in the cochlea, and they configure the sound waves in the scala vestibuli $P_{SV}(x, t)$ and scala tympani $P_{ST}(x, t)$.

Here, it should be noted that the pressure level of the odd mode, $P_{ODD}(x, t)$, reduced with the position $x$, and finally, it vanished. Instead, the displacement of the basilar membrane grew largely in the cochlea. This fact indicates that the odd mode $P_{ODD}(x, t)$ played an important role in exciting the traveling wave on the basilar membrane. However, more importantly, the even mode $P_{EVEN}(x, t)$ was not independent of the traveling wave, and it was deeply related to the odd mode $P_{ODD}(x, t)$. The even sound wave mode excited at the cochlea base ($x = 0$ mm) traveled along the cochlea, and it was reflected perfectly at the cochlea apex ($x = 35$ mm) with a fixed-end reflection condition. As a result, the even-mode sound wave generated a standing wave in the cochlea. Therefore, the pressure level of the even mode $P_{EVEN}(x, t)$ became a function of the cochlea length. Under such behaviors of the even and odd-mode sound waves, $P_{ODD}(0$ mm$, t)$ should be equal to $P_{EVEN}(0$ mm$, t)$ to meet the condition $P_{ST}(0$ mm$, t) = 0$ at the base of the scala tympani.

### 2.4. Validity of Even and Odd Mode Approaches in Other Frequencies

In the previous section, we introduced how the cochlea works acoustically at 5000 Hz based on the even and odd-mode sound waves, $P_{EVEN}(x, t)$ and $P_{ODD}(x, t)$. In this section, we describe the verification of the validity of this approach at frequencies of 2000 Hz, 5000 Hz, and 10,600 Hz.

Figure 3a shows the $P_{SV}(x, t)$, $P_{ST}(x, t)$, $P_{EVEN}(x, t)$, and $P_{ODD}(x, t)$, and the displacement of the basilar membrane when the cochlea was excited by a continuous sinusoidal plane wave with a sound pressure level of 1 Pa and a frequency of 2000 Hz. When the velocity of the sound wave traveling in the perilymph was assumed to be 1520 m/s, the wavelength of the sound wave at 2000 Hz was estimated to be 0.76 m. Since the wavelength was long enough at this frequency compared to the total length of the cochlea of 35 mm, the pressure level of the even mode $P_{EVEN}(x, t)$ became almost constant anywhere in the cochlea, while the waveform of the odd mode $P_{ODD}(x, t)$ vibrated at the position from 0 mm to 20 mm, and, finally, it vanished beyond 20 mm. At the same time, the displacement of the basilar membrane was greatly generated. Here, the sound pressure levels of the even mode and odd mode at the base of the cochlea could be read as $P_{EVEN}(0\ mm, t) = +0.12$ Pa and $P_{ODD}(0\ mm, t) = +0.14$ Pa, respectively. When we applied the even and odd-mode approach to the model, as explained in the previous section, $P_{ST}(0\ mm, t)$ was $-0.02$ Pa, not 0 Pa. The reason for this is that the traveling wave excited on the basilar membrane did not converge sufficiently at the apex of the cochlea, and we considered that it was reflected slightly at the end of the basilar membrane. The reflected traveling wave was transformed into the sound wave again and traveled back to the cochlea base. However, if we increase the simulation time, $P_{ST}(0\ mm, t)$ will converge to zero.

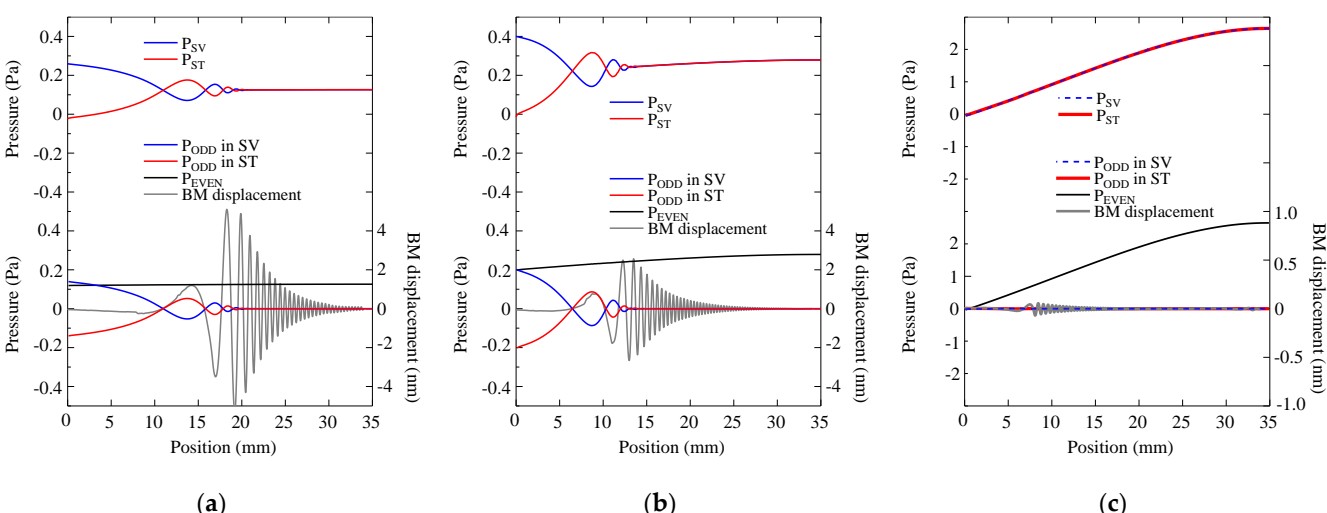

**Figure 3.** The upper graph shows the sound pressure levels in the scala vestibuli (in blue) and scala tympani (in red). The lower graph presents the sound pressure levels of the even mode (in black) and odd modes in the scala vestibuli (in blue) and scala tympani (in red), and the displacement of the basilar membrane (in light gray) when a continuous sinusoidal plane wave with a pressure level of 1 Pa was excited at the oval window. The horizontal axis shows the position in the cochlea. The observation times $t = 30.096$ ms, 38.840 ms, and 21.090 ms were chosen by trial and error so that the waveform swing $P_{EVEN}$ became the maximum after the waveform reached a steady state. (**a**) $f = 2000$ Hz, $t = 30.096$ ms. (**b**) $f = 5000$ Hz, $t = 38.840$ ms. (**c**) $f = 10,600$ Hz, $t = 21.090$ ms.

Next, Figure 3b shows the simulation results when the oval window was excited at 5000 Hz, which was discussed in the previous section. The graph was redrawn in the same style as the other graphs in Figure 3 to facilitate the comparison. The sound pressure levels of the even and odd modes at the cochlea base were $P_{EVEN}(0\ mm, t) = +0.2$ Pa and $P_{ODD}(0\ mm, t) = +0.2$ Pa, respectively, and $P_{SV}(0\ mm, t) = +0.4$ Pa and $P_{ST}(0\ mm, t) = 0.0$ Pa

were obtained. To obtain such an ideal result, it is necessary that the traveling wave excited by the odd mode does not cause reflection at the end of the basilar membrane.

Finally, Figure 3c shows the simulation results when the oval window was excited at 10,600 Hz. This is a special case, because the sound pressures on the scala vestibuli and scala tympani were almost equal, so there was no difference in sound pressure between them. As a result, the sound pressure in the odd mode was almost zero everywhere in the cochlea. As can be read from the figure, the displacement of the basement membrane reduced significantly and was negligible compared to other cases, as shown in Figure 3a,b. On the other hand, regarding the amplitude of the even mode, a standing wave was generated in the cochlea so that the pressure level became zero at the cochlea's base ($x = 0.0$ mm) and maximum at the apex ($x = 35$ mm). In other words, this specific phenomenon occurred only when the cochlea was excited by a sound wave whose quarter wavelength was coincident with the cochlea length of 35 mm. It should be noted that the vertical axis of this figure was drawn on a different scale to that of the other graphs for clarity. In addition, the reason that the even mode formed a standing wave with a sound pressure exceeding 2 Pa at the cochlea apex is that the shape of the cochlea is tapered.

We explained in [25] that even and odd sound wave modes existed in the cochlea, and when a continuous sinusoidal plane wave was excited at the oval window, even and odd modes were generated in the cochlea at first, and then the odd mode generated the traveling wave on the basilar membrane. In the previous discussion, the validity of the even and odd mode approach was confirmed at 2000 Hz, 5000 Hz, and 10,600 Hz. However, Figure 3 shows the displacement of the BM at a certain time. By checking the envelope of the displacement in the time domain, the maximum displacement of the BM in the cochlea was determined. Based on this procedure, the simulation was carried out from 500 Hz to 20,000 Hz, and the result is summarized in Figure 4. It can be observed from the figure that the displacement of the basilar membrane tended to be larger at the lower frequency range. Generally, this is because an elastic membrane vibrates more at a lower frequency when a sound wave with the same pressure level is set for excitation. However, looking at the response at 10,600 Hz, the displacement of the basilar membrane was suddenly reduced. This means that the cochlea sensitivity is weakened at this frequency. The reason for this is that the cochlea length of 35 mm became a quarter wavelength at 10,600 Hz, and then the sound level of the even mode became zero at the cochlea base; that is, $P_{EVEN}(0 \text{ mm}, t) = 0$. According to the previous section, the following conditions were required at the healthy cochlea base:

$$P_{SV}(0 \text{ mm}, t) = P_{EVEN}(0 \text{ mm}, t) + P_{ODD}(0 \text{ mm}, t) \tag{9}$$

$$P_{ST}(0 \text{ mm}, t) = P_{EVEN}(0 \text{ mm}, t) - P_{ODD}(0 \text{ mm}, t) = 0 \tag{10}$$

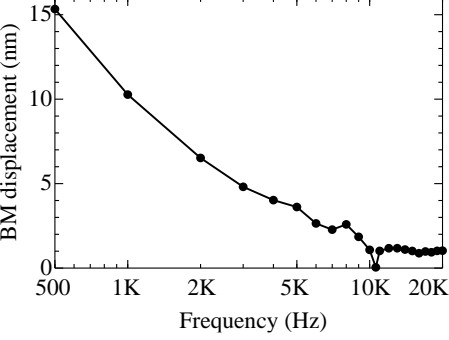

**Figure 4.** Frequency dependence of the displacement of the basilar membrane when a continuous sinusoidal plane wave of 1 Pa was excited at the oval window.

In this case, $P_{ODD}(0 \text{ mm}, t)$ should be zero, and, accordingly, the pressure level at the oval window also should be zero; that is, $P_{SV}(0 \text{ mm}, t) = 0$, which means that the oval window works as a reflector with the free-end reflection condition. In other words, the

input impedance of the cochlea becomes $0\,\text{Pa}\cdot\text{s}/\text{m}^3$, and the cochlea does not accept any sound waves coming into the cochlea [25]. Even though our simulation result cannot be compared to the loudness curve reported in [33] directly because our simulation results do not consider the frequency characteristics of the outer and middle ears and the nonlinear response of the cochlea amplifier caused by the protein motor *Prestin*, the loudness curve in [33] also showed hearing deterioration in that area. The simulated hearing loss around 10,600 Hz seems to be more severe than the measured results in [33]. This is because an ideal hard boundary condition was given to the side walls of the scala vestibule and scala tympani, and the leakage of the sound waves through the temporal bone was ignored in the simulation. We expect that, if the leakage of the sound waves was considered as demonstrated in [34,35], the severe hearing loss in our simulation would be milder and closer to the actual hearing properties of humans.

### 3. Hearing Loss by Round Window Atresia

#### 3.1. Round Window Atresia

The membrane of the round window has a three-layer architecture whose middle core of the connective tissue contains collagen fibers, fibroblasts, and other elastic fibers to maintain the structure of the round window [31,32]. The shape of the round window is mostly elliptical [30], and it is said that the round window reaches adult size already in the early stages of fetal development [36]. According to the study of Jain et al., the length of the minor axis of the round window is 0.51–1.27 mm, with an average value of $0.69 \pm 0.25$ mm, and the length of the major axis is 0.51–2.04 mm, with an average value of $1.16 \pm 0.47$ mm [27]. Generally, the adult round window is thicker at the edges than at the center, with an average thickness of approximately 70 μm [31,32].

There are only a few types of diseases related to the round window. In most cases, symptoms such as dizziness, tinnitus, ear fullness, and hearing loss are caused by perilymph leakage due to damage to the round window [37]. Round window atresia is one of the few diseases associated with the round window, but it is similar to the symptoms of diseases such as otosclerosis, so it is difficult to identify it in the initial diagnosis without high-resolution image diagnosis [38]. However, it is true that round window atresia causes serious hearing loss [38–41], and this fact indicates that the mobility of the round window is essential to maintaining healthy hearing.

In this section, we describe the design of a round window atresia model in which the mobility of the round window is defined by the Young's modulus, and discuss how the reflection properties change at the round window and affect the excitation of the traveling waves on the basilar membrane. Finally, the obtained results are compared with the measured reports from the medical side to show the validity of our study.

#### 3.2. Reflection Properties of the Round Window with Round Window Atresia

In order to simulate the acoustic properties of the round window with round window atresia, a rectangular acoustic tube model was designed, as shown in Figure 5. An *x*-axis was defined along the black line from *input a* to *input a′*, and an origin was given on the plane *input a*. The model was configured by two regions: one was an air-filled region (0 mm < *x* < 3 mm, assuming the middle ear cavity), and the other was a perilymph-filled region (3 mm < *x* < 38 mm, assuming the scale tympani), which were separated by an elliptical-shaped elastic membrane (assuming the round window). The tapered perilymph-filled region (length of $L_s = 35$ mm and cross-sections of $W_{bs} = 1.2$ mm and $H_{bs} = 1.235$ mm at $x = 3$ mm, and $W_{as} = 0.7$ mm and $H_{as} = 0.745$ mm at $x = 38$ mm) and the air-filled region (dimensions of $L_a = 3$ mm, $W_a = 1.2$ mm, and $H_a = 1.253$ mm) faced each other across the round window. Since the actual round window has an elliptical shape, the lengths of the major and minor axes of the round window were set as $W_r = 1.12$ mm and $H_r = 0.69$ mm. The membrane of the round window was treated as an elastic material with a thickness of $T_r = 70$ μm, a Young's modulus of $E$, a Poisson ratio of 0.49, and a mass density of 1200 $\text{kg}/\text{m}^3$, whose peripheral part was fixed to a partition separating these two regions.

An ideal hard boundary condition, that is, the condition that the normal component of the particle velocity is always zero on the wall surface, was applied to the inner walls and the partition. An observation line was set on the *x*-axis to evaluate the sound pressure level in the air-filled and perilymph-filled regions. Then, a continuous sinusoidal plane wave with a sound pressure level of 1 Pa and a frequency of 5000 Hz was excited from *Input a* of the perilymph-filled region.

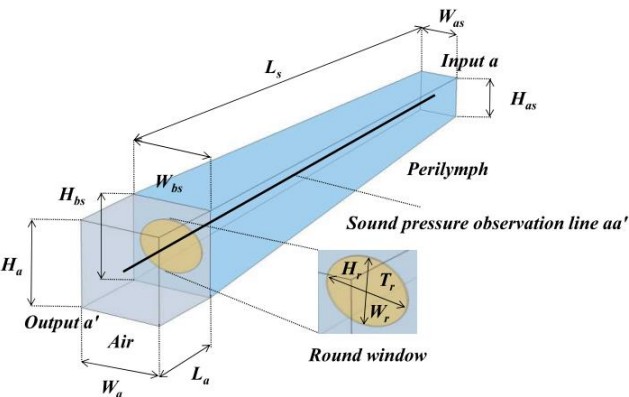

**Figure 5.** Rectangular acoustic tube model configured by an air-filled region and a perilymph-filled region, both of which were connected by an elliptical-shaped elastic membrane. The dimensions were $L_s$ = 35 mm, $W_{bs}$ = 1.2 mm, $H_{bs}$ = 1.235 mm, $W_{as}$ = 0.7 mm, $H_{as}$ = 0.745 mm, $L_a$ = 3 mm, $W_a$ = 1.2 mm, $H_a$ = 1.253 mm, $W_r$ = 1.12 mm, $H_r$ = 0.69 mm, and $T_r$ = 70 μm.

In order to demonstrate three stages of round window atresia, the mobility of the round window membrane was defined by the Young's modulus *E* as follows.

- $E$ = 1 MPa (normal mobility—healthy ear);
- $E$ = 100 MPa (lack of mobility to some extent—mild round window atresia);
- $E$ = 10 GPa (complete lack of mobility—severe round window atresia).

Figure 6 presents the simulation results for these cases. The horizontal axis in the figure shows the position on the observation line; that is, the position from 0 mm to 3 mm corresponds to the air-filled region, and that from 3 mm to 38 mm corresponds to the perilymph-filled region. The round window was located at 3 mm, and the excitation plain was set at 38 mm.

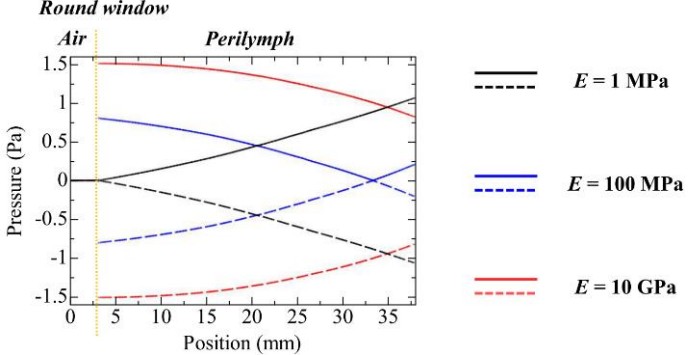

**Figure 6.** Sound pressure levels evaluated on sound observation line *aa'* in Figure 5. Assuming three stages of round window atresia, the Young's modulus of the round window membrane was set as $E$ = 1 MPa (a healthy ear, in black), $E$ = 100 MPa (an ear with mild round window atresia, in blue), and $E$ = 10 GPa (an ear with severe round window atresia, in red). The position from 0 mm to 3 mm corresponds to the air-filled middle ear region, and the position from 3 mm to 38 mm corresponds to the perilymph-filled scala tympani region. A sinusoidal plane wave with a sound pressure level of 1 Pa and a frequency of 5000 Hz was set for the excitation.

The solid and dashed black curves in the figure represent the maximum and minimum sound pressure levels, respectively. When the round window was in the healthy state ($E = 1$ MPa), the actual sound pressure level in the tube would change over time between these two curves. Contrarily, as shown in the graph, the sound pressure in the air-filled region always became zero approximately. This means that, although the sound wave was excited in the perilymph-filled region, it did not enter the air-filled region beyond the round window due to the large impedance mismatch between the air and perilymph; that is, the round window membrane provided a free-end reflection condition to the sound waves traveling in the perilymph region and caused a standing wave in that region.

Next is the critical case when the mobility of the round window is completely lost due to severe round window atresia ($E = 10$ GPa), and the simulation results are shown in red in the same figure. Similar to the upper case, the sound wave was reflected at the round window, and a standing wave was generated in the perilymph region. However, different from the previous case, a fixed-end reflection condition was given to the sound wave traveling in the perilymph-filled region, since the round window membrane did not move at all. In fact, the maximum swing of the sound pressure level reached 1.5 Pa at the round window.

Finally, we present a case when the mobility of the round window is lost to some extent due to mild round window atresia ($E = 100$ MPa). As indicated in blue in the figure, the sound pressure responses present an intermediate property between the free-end reflection shown in black and the fixed-end reflection in red. This indicates that the mobility of the round window can change by tuning the Young's modulus of the round window membrane, corresponding to the stage of round window atresia. In the next section, the cochlea model with round window atresia is discussed.

Figure 7 shows the displacements of the round window for three stages of round window atresia. When $E = 1$ MPa (a healthy ear), the round window could move freely without restriction, and the largest displacement was obtained. On the other hand, when $E = 10$ GPa (an ear with severe round window atresia), the flexibility of the round window was completely lost. This means that the round window does not move and it works as a fixed-end reflector to the sound waves traveling in the perilymph.

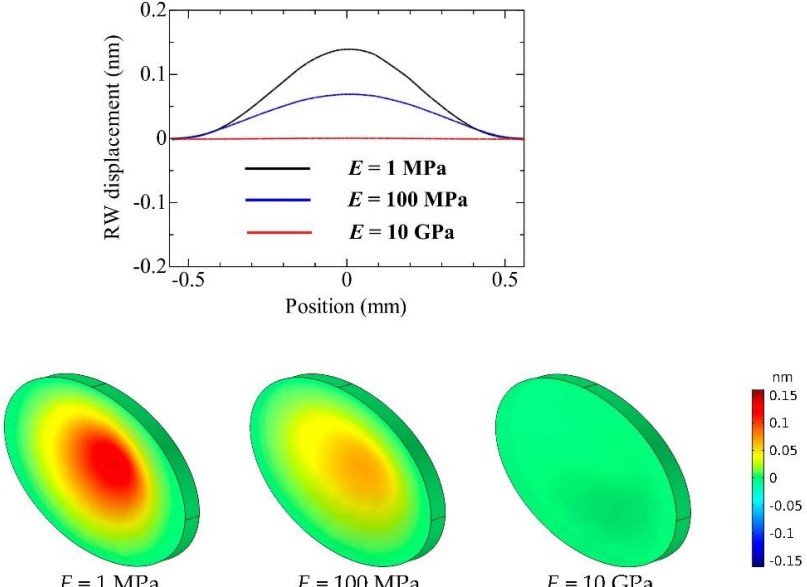

**Figure 7.** Displacements of the round window for three stages of round window atresia. The upper graph shows the maximum displacement of the elliptical round window membrane evaluated on its long axis. Position 0 mm corresponds to the center of the round window membrane. The lower ones also show the displacement of the round window membrane expressed by colors. A sinusoidal plane wave with a sound pressure level of 1 Pa and a frequency of 5000 Hz was set for the excitation.

### 3.3. Demonstration of Round Window Atresia

A new cochlea model with round window atresia was designed by using the straight-tapered cochlea model shown in Figure 1. The configuration, physical dimensions, and material parameters were identical to those in the original cochlea model, except for the Young's modulus of the round window membrane. As mentioned in the previous section, the Young's modulus was defined as $E = 1$ MPa for the healthy ear, $E = 100$ MPa for mild round window atresia, and $E = 1$ GPa for severe round window atresia.

As shown in Section 2.4, the sound pressure levels in the scala vestibuli and scala tympani, and the displacement of the basilar membrane, were demonstrated by using the healthy cochlea model without round window atresia. Additionally, the acoustic behavior of the cochlea was explained based on the even and odd mode approach. By using the same procedure, these parameters were calculated for the cochlea model with severe round window atresia ($E = 1$ GPa). The simulation results are summarized in Figure 8. It can be seen from the figure that the sound pressure levels of the even mode were $P_{EVEN}(0 \text{ mm}, t) = 1.1$ Pa at the base and $P_{EVEN}(35 \text{ mm}, t) = 1.5$ Pa at the apex, while in the case of the healthy ear, the sound pressure levels of the even mode were $P_{EVEN}(0 \text{ mm}, t) = 0.20$ Pa at the base and $P_{EVEN}(35 \text{ mm}, t) = 0.26$ Pa at the apex, as presented in Figure 3b. This means that the even mode becomes dominant when round window atresia is becoming severe, and the excitation of the odd mode is weakened. As a result, the displacement of the basilar membrane, which is excited by the odd mode, is largely reduced and the hearing loss will become more severe.

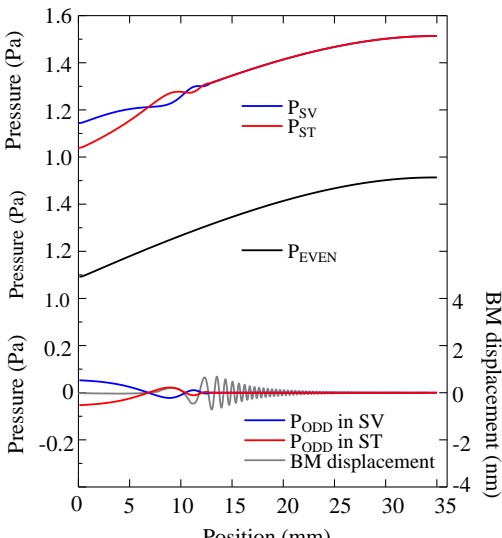

**Figure 8.** Analysis of an ear with severe round window atresia ($E = 10$ GPa). The upper graph shows the sound pressure levels in the scala vestibuli (in blue) and scala tympani (in red). The middle graph shows the sound pressure level of the even mode (in black). The lower graph presents the sound pressure levels of the odd modes in the scala vestibuli (in blue) and scala tympani (in red), and the displacement of the basilar membrane (in light gray) when a sinusoidal plane wave with a pressure level of 1 Pa and a frequency of 5000 Hz was excited at the oval window. The horizontal axis shows the position in the cochlea.

Next, the frequency dependence of the displacement of the basilar membrane was studied, and the results are summarized in Figure 9. The black curve in the figure represents the case of the healthy ear ($E = 1$ MPa), the blue curve represents the case of mild round window atresia ($E = 100$ MPa), and the red curve represents the case of severe round window atresia ($E = 10$ GPa). These were obtained by checking the envelope of the displacement in the time domain and finding the maximum displacement of the basilar membrane at each frequency in the same manner as in Figure 4. As explained in Figure 4, the displacement of the basilar membrane tended to be larger at a lower frequency when a sound wave with the

same pressure level was set for excitation. This is true for a healthy ear. However, once the mobility of the round window is restricted due to round window atresia, the displacement of the basilar membrane is significantly weakened, especially at a lower frequency. This suggests that round window atresia may cause severe hearing loss in a lower frequency range, as shown in Figure 9.

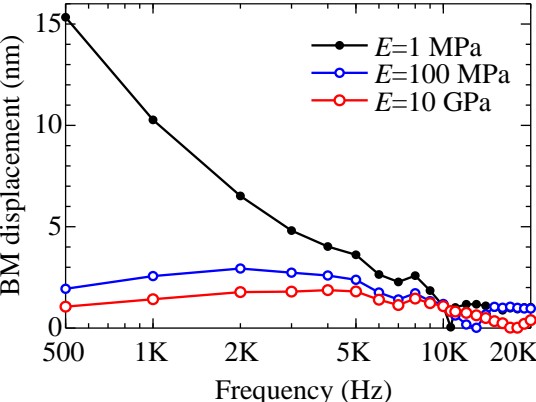

**Figure 9.** Frequency response of the maximum displacement of the basilar membrane when a sinusoidal plane wave with a pressure level of 1 Pa was excited at the oval window. Displacements of the basilar membrane for a healthy ear (*E* = 1 MPa, in black), an ear with mild round window atresia (*E* = 100 MPa, in blue), and an ear with severe round window atresia (*E* = 10 GPa, in red) are presented, where *E* stands for the Young's modulus of the round window membrane.

*3.4. Hearing with Round Window Atresia*

All of the studies presented above merely dealt with the acoustic properties of the cochlea itself when a sinusoidal plane wave with a pressure level of 1 Pa was set for the oval window. However, in real life, we catch the vibration of sounds using the pinna and detect it with the cochlea. This means that hearing ability should be evaluated as a whole auditory system, including the frequency characteristics of the outer, middle, and inner ears. As shown in the upper graph in Figure 10, the frequency characteristics of the outer and middle ears are reported [42,43]. Based on this information, the sound pressure levels reached in the oval window were calculated when the pinna was excited by a sinusoidal plane wave with a pressure level of 1 Pa. Then, the displacement of the basilar membrane was estimated by exciting the oval window with the sound pressure level reached there.

By taking the frequency characteristics of the outer and middle ears into consideration, the displacement of the basilar membrane was obtained, as shown by the middle graph in Figure 10. Though the graph is not very smooth because the original measured data of the outer and middle ears in [42,43] were discrete and reported up to 10,000 Hz, the graph indicates that the sensitivity of the human auditory system showed local maximums at 250 Hz and from 2000 Hz to 3000 Hz. Following that, the unit "Pa" of the middle graph in Figure 10 was converted to "dB" and normalized by the value of 1000 Hz, and the normalized hearing sensitivity of the human auditory system was obtained, as shown by the lower graph in Figure 10. Strictly, this result cannot be quantitatively compared to the human audiogram because the nonlinear effect of the cochlea amplifier provided by the protein motor *Prestin* is not yet considered. However, the lower graph in Figure 10 is considered to be useful when the normalized hearing sensitivity of the human auditory system is discussed qualitatively.

Based on this approach, the normalized sensitivity of the human auditory system with round window atresia was estimated. As in Section 3.2, the symptoms of round window atresia were classified into three stages: a healthy ear without round window atresia (*E* = 1 MPa, in black), an ear with mild round window atresia (*E* = 100 MPa, in blue), and an ear with severe round window atresia (*E* = 10 GPa, in red), where *E* stands for the Young's modulus of the round window membrane. The simulated results are shown in

Figure 11. Looking at the sensitivity of the healthy ear, the hearing sensitivity reached the maximum around 3000 Hz and began to deteriorate above 5000 Hz.

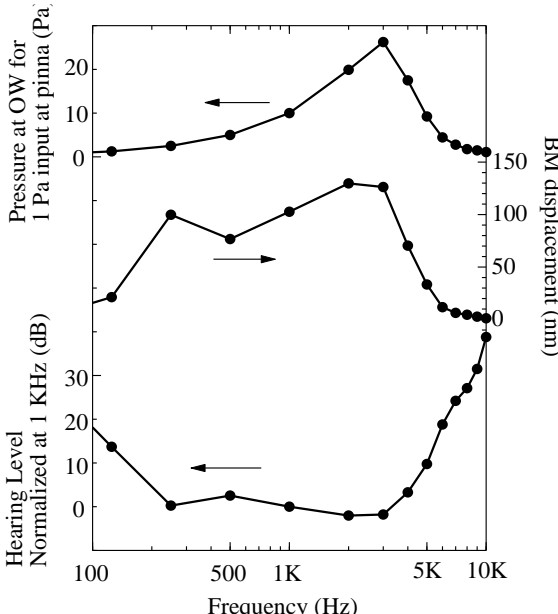

**Figure 10.** Estimation of the hearing level of the whole auditory system, including the properties of the outer, middle, and inner ears, when the ear is healthy. The upper graph shows the sound pressure level at the oval window when a sinusoidal plane wave with a pressure level of 1 Pa was excited at the pinna [42,43]. The middle graph shows the displacement of the basilar membrane when the pressure levels estimated above were used for sound wave excitation at the oval window. The lower graph presents the hearing level of the whole human auditory system. The result is expressed in "dB", normalized at 1000 Hz, and redrawn so as to be compared to the general audiogram.

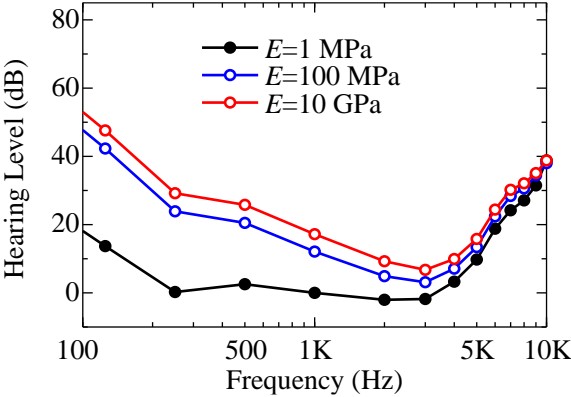

**Figure 11.** Hearing level of the whole auditory system estimated for a healthy ear (*E* = 1 MPa, in black), an ear with mild round window atresia (*E* = 100 MPa, in blue), and an ear with severe round window atresia (*E* = 10 GPa, in red), where *E* stands for the Young's modulus of the round window membrane.

Regarding round window atresia, Mansour et al. classified the level of round window atresia into five stages, ranging from RW-I, where the CT value decreases only at the margins of the round window, to RW-V, which has a wide range of lesions [44]. According to his classification, our demonstrations of setting the Young's modulus of the round window membrane to *E* = 100 MPa and *E* = 10 GPa are both classified into the RW-IV stage, in which round window atresia is ossified. From this point of view, we would like to verify Sonoda's report concerning the measured results of the pure-tone audiometry

examined for patients who had round window atresia [38]. In the study, he compared the hearing ability of subjects with and without round window atresia, and reported that hearing was deteriorated by about 10 dB to 20 dB below 4000 Hz due to round window atresia. However, the reasons for this were not clearly described in his report.

Now, let us revisit our simulation results presented in Figure 11. Comparing the sensitivity of the healthy ear and the ears with round window atresia ($E = 100$ MPa or $E = 10$ GPa), the sensitivity was deteriorated by 10 dB to 20 dB between 1000 Hz and 4000 Hz, and 20 dB to 30 dB between 100 Hz and 1000 Hz. In addition, the deterioration of hearing became more severe at a lower frequency. These results show good agreement with Sonoda's report, and we believe that our approach is useful in elucidating the mechanism of round window atresia.

## 4. Conclusions

In this paper, we designed a straight-tapered cochlea model, including the compressible perilymph-filled scala vestibuli, scala tympani, and helicotrema. The basilar membrane and the round window membrane were modeled carefully to discuss how round window atresia affects our hearing. For this study, we combined two modules for the simulation software COMSOL Multiphysics. One was a pressure acoustics and thermoviscous acoustics module for acoustics, and the other was a structural mechanics module for the analysis of elastic materials. Though the basic idea was introduced in [25], to improve readers' understanding, we demonstrated the detailed acoustic aspects of the cochlea based on the even and odd mode approach, and explained the following important points:

(1) The sound waves traveling in the cochlea were classified into the even and odd symmetric modes and were expressed by the sum of these modes;

(2) The odd mode excited the displacement of the basilar membrane and generated the Békésy's traveling wave on the membrane;

(3) The even mode generated a standing wave in the cochlea due to a fixed-end reflection at the cochlea apex;

(4) The acoustic properties of the cochlea were determined by the contributions of the even and odd modes.

Following that, we analyzed the cochlea model with round window atresia by changing the Young's modulus of the round window membrane, and obtained the following new facts:

(5) When the Young's modulus $E$ of the round window membrane is normal (e.g., a healthy ear with $E = 1$ MPa), the round window membrane provides a free-end reflection condition against the sound waves traveling in the scala tympani heading to the round window;

(6) When the Young's modulus $E$ of the round window membrane is higher (e.g., an ear with severe round window atresia $E = 10$ GPa), the round window membrane provides a fixed-end reflection condition against the sound waves traveling in the scala tympani heading to the round window;

(7) It is reported from the clinical medicine perspective that patients who have round window atresia tend to have their hearing ability degraded by 10 dB to 20 dB below 4000 Hz. Our simulation results show good agreement with the reported symptoms, and this ensures that our approach to round window atresia is correct.

Finally, we would like to emphasize that such simulation results were obtained only when the perilymph was assumed to be compressible and the sound waves in the cochlea were treated as a compression wave.

**Author Contributions:** Conceptualization, W.H. and Y.H.; software, W.H.; writing—original draft preparation, W.H.; writing—review and editing, Y.H.; supervision, Y.H.; project administration, Y.H.; funding acquisition, Y.H. All authors have read and agreed to the published version of the manuscript.

**Funding:** This research was financially supported by the Kansai University Fund for the Promotion and Enhancement of Education and Research, 2020–2022. "Engineering study on auditory mechanism for innovative development of clinical medicine".

**Institutional Review Board Statement:** Not applicable.

**Informed Consent Statement:** Not applicable.

**Data Availability Statement:** The data that support the findings of this study are available from the corresponding author upon reasonable request.

**Conflicts of Interest:** The authors declare no conflict of interest.

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
