# Peer review of "Simulation-Based Study on Round Window Atresia by Using a Straight Cochlea Model with Compressible Perilymph"

_acoustics, doi:10.3390/acoustics4020021_

Round 1
Reviewer 1 Report
The authors proposed a numerical model for cochlea considering compressible perilymph and examined the effects of round window stiffness on human hearing. The results and conclusions were clearly presented. However, here are some concerns on the novelty and numerical analysis of this study:
- As the authors said, the major contribution of this study on the modeling is treating the perilymph as a compressible fluid and some previous papers have already done that. Then, what would be the major differences and novelty between this paper and the previous studies? Does your model provide extra insights into the mechanisms of round window atresia by using this model?
- In the 2.2 section, the authors used the "sound hard boundary" for the walls in the simulation. What's the justification here? Is it because the inner ear is surrounded by temporal bone? How is the volume of the current domain compared to the human ear?
- The authors should mention more details on how to set up the finite-element-analysis modeling, such as the mesh size, time step for time-dependent simulation, and computation time. I'd also recommend the authors list a table showing all the materials used in the modeling and their properties. That is a common way in FEM paper.
- In the 2.3 section, the authors should also clarify the excitation waveform on the oval window. Is it a continuous sinusoidal wave or pulse wave?
- Why is the maximum swing picked at t=38.84 ms? What does the time mean here? Time to the steady-state? Why only show maximum and minimum swings here? Any physical interpretations? Please clarify.
- I was a little confused when I read the 3.2 section. I understand the authors intend to show the reflection properties of the round window, but why use a totally different cylindrical tube model and then continue your results based on the earlier model? Is there any specific reason?
- In line 79, "recommend to do" is not correct in grammar.
- Would it be more appropriate to change the title on the y-axis in Figure 10 from "Sound Pressure Level (dB)" to be "Hearing level (dB)"?
Author Response
Dear Mr./Ms Reviewer1,
Thank you very much for your valuable time for reviewing of our manuscript.
I'm attaching the reply letter.
I wish you my best regards,
Yasushi Horii

Reviewer 2 Report
In this research, a straight-tapered cochlea model and the effects of the atresia of the round window on the hearing have been studied. The article is well-organized and has an adequate impact. The problem is timely and interesting. The title matches the content and the abstract is self-reliant and complete. I recommend the publication of the manuscript after the following improvements is done
1) There are several types as well as grammatical errors which make the text difficult to understand on many occasions.
2) The abstract is briefly organized. Authors are encouraged to add more details on the Abstract section.
3) Although the title, materials and methods of the present paper are very interesting, however the main novelties of it are not clear for the reviewer. Authors are encouraged to add more comments on the novelties and main contributions of the present paper in Abstract and last paragraph of Introduction section.
4) The authors need to clarify boundary conditions
5) Figure 2 is cluttered (legend and the title of the axes….)
6) The current references are appropriate. However, the reference list is short. Consequently, some new references about the research subject should be added to the reference list and cited in the text. For instance,
- Xue, Lin, et al. "Research on coupling effects of actuator and round window membrane on reverse stimulation of human cochlea." Proceedings of the Institution of Mechanical Engineers, Part H: Journal of Engineering in Medicine 235.4 (2021): 447-458.
- Thongchom, C., Saffari, P.R., Refahati, N., Saffari, P.R., Pourbashash, H., Sirimontree, S. and Keawsawasvong, S., 2022. An analytical study of sound transmission loss of functionally graded sandwich cylindrical nanoshell integrated with piezoelectric layers. Scientific Reports, 12(1), pp.1-16.
- Zhang, Jing, et al. "Finite element analysis of round-window stimulation of the cochlea in patients with stapedial otosclerosis." The Journal of the Acoustical Society of America 146.6 (2019): 4122-4130.
- Thongchom, C., Jearsiripongkul, T., Refahati, N., Roudgar Saffari, P., Roodgar Saffari, P., Sirimontree, S. and Keawsawasvong, S., 2022. Sound Transmission Loss of a Honeycomb Sandwich Cylindrical Shell with Functionally Graded Porous Layers. Buildings, 12(2), p.151.
Author Response
Dear Mr./Ms Reviewer2,
Thank you very much for your valuable time for reviewing of our manuscript.
I'm attaching the reply letter.
I wish you my best regards,
Yasushi Horii
Round 2
Reviewer 2 Report
Corrections were well done.